# LEVERAGING LLM HIDDEN-STATE TRAJECTORIES FOR LABEL-FREE SAMPLER ADAPTATION

## ABSTRACT

In-context learning is an emergent capability of large language models that solves unseen tasks by conditioning on a few demonstrations without updating parameters. ICL performance hinges on how the sampler selects demonstrations. In deployment, distribution shift is common and target labels are scarce due to privacy and cost, making cross-domain retrieval both important and challenging. Yet sampler behavior under such shifts remains underexplored. Our analysis shows that the relevance–performance and diversity–performance relationships vary by domain, and that characterizing this trade-off in the target domain is essential for sampler generalization in cross-domain settings. However, without labels it is impossible to evaluate this trade-off, so we leverage the geometry of LLM embedding trajectories (the Chain of Embedding, CoE), defined as the sequence of hidden states across layers, as a label-free signal. We show that broader exploration of the trajectory, measured by PCA hull volume, flags domains where diversity correlates positively with performance. Building on this insight, we propose LRPG (Latent Reasoning Path Guidance), a lightweight method that decides whether to increase diversity based on CoE statistics without requiring labels. Across diverse benchmarks, existing samplers suffer large drops in cross-domain settings, whereas LRPG consistently improves target-domain ICL performance and composes orthogonally with existing sampler designs.

## 1 INTRODUCTION

Large language models (LLMs) at sufficient scale exhibit *emergent abilities*, which are capabilities absent in smaller models but appearing unexpectedly as model size increases, such as logical reasoning or few-shot generalization (Wei et al., 2022). One of the most notable examples is *in-context learning* (ICL), where an LLM solves a new task by conditioning on a test instance together with a handful of input–output demonstrations, without any gradient updates to model parameters (Brown et al., 2020). This ability allows practitioners to harness LLM reasoning capacity even in low-resource scenarios.

One central challenge, however, is that the effectiveness of ICL depends heavily on which demonstrations are included in the context. The choice of examples can drastically alter downstream performance, and thus selecting demonstrations has emerged as a key research problem. Early *learning-free* methods ranked candidates using lexical overlap or embedding similarity (Liu et al., 2021), while more recent *learning-based* samplers train retrieval models that balance relevance and diversity (Rubin et al., 2022; Ye et al., 2023; Wang et al., 2024b;c). Although these approaches achieve strong in-domain performance, their behavior under cross-domain shifts remains poorly understood. Tackling this cross-domain setting is essential for designing *generalizable samplers*, which can operate effectively when the target domain offers few or no labels, a situation frequently encountered in practical pipelines such as commonsense reasoning, or dialogue systems.

To systematically address this problem, we pose two central questions. First, we ask whether the relevance–diversity trade-off with respect to performance is consistent across datasets (Q1). This question is important because prior work has largely optimized samplers by tuning this trade-off in-domain, but it is unclear whether the same balance holds once we move across domains. Our analysis shows that this balance is not universal since some domains benefit from highly relevant demonstrations while others improve with more diverse examples. Second, we examine whether

such domain-dependent trade-off patterns persist for learning-based samplers (Q2). This matters because learning-based retrievers are often assumed to be more adaptive and robust than heuristic methods, but if they still inherit domain-specific biases, then their cross-domain generalization remains fundamentally limited. We find that learning-based retrievers inherit these domain-dependent trends, and their performance is primarily governed by the target domain's diversity–performance trade-off. Together, Q1 and Q2 highlight a fundamental limitation of cross-domain ICL. Without access to labels in the target domain, it is difficult to determine the appropriate relevance–diversity trade-off for effective retrieval.

This naturally raises a third question: how can one infer the relevance–performance and diversity–performance relationship in a target domain without labels (Q3)? To approach this challenge, we draw on the *Chain-of-Embedding* (CoE) (Ren et al., 2022; Wang et al., 2024d;e), the sequence of hidden states across layers produced during inference. We regard CoE as a latent reasoning path that reflects how internal representations evolve within the model. Our analysis shows that the geometry of this trajectory, measured for instance by the volume of its principal-component hull, provides a reliable label-free signal of domain characteristics. In particular, domains with broader embedding trajectories tend to favor diversity, whereas domains with narrower trajectories favor relevance. This observation suggests that CoE can be used to uncover these trade-offs in a target domain without requiring labeled data.

Building on these insights, we propose LRPG (Latent Reasoning Path Guidance), a lightweight adaptation method that can be applied orthogonally to existing learning-based samplers. LRPG leverages CoE geometry to infer whether a target domain benefits more from relevance or diversity, based on correlations observed across multiple source domains. During training or inference, LRPG uses the target domain's reasoning path as a label-free signal to guide the adjustment of the sampler's diversity component, thereby improving retrieval without any target-domain labels. To evaluate this idea, we design a new experimental setting that extends standard benchmarks with explicit cross-domain splits. Across diverse tasks, LRPG consistently mitigates the severe performance drops of existing samplers in cross-domain conditions and yields notable improvements in ICL accuracy.

Our contributions are summarized as follows:

- **Cross-domain analysis of ICL samplers.** We conduct the first systematic study of relevance–diversity–performance correlations across domains, showing that these patterns are highly dependent on the target domain.

- **Embedding trajectory geometry as a label-free signal.** We show that the Chain-of-Embedding captures latent reasoning paths whose geometry provides a reliable indicator of whether a domain benefits more from relevance or from diversity.

- **Latent Reasoning Path Guidance (LRPG).** We propose a lightweight, label-free adaptation method that leverages CoE geometry as a signal to infer the relevance–diversity trade-off in a target domain. By using the target domain's reasoning path to guide the direction of diversity adjustment during training or sampling, our approach can be seamlessly integrated into existing samplers. This provides orthogonal improvements to prior methods and consistently enhances cross-domain ICL performance.

## 2 PRELIMINARIES

### 2.1 IN-CONTEXT LEARNING

Let $\mathcal{X}$ denote the input space and $\mathcal{Y}$ the output space. Each input $x \in \mathcal{X}$ is associated with a ground-truth output $y \in \mathcal{Y}$. A large language model with parameters $\theta$ defines a conditional distribution $P_\theta(y \mid \text{prompt})$ over outputs given a textual prompt. In $N$-shot *in-context learning* (ICL), a large language model receives a prompt as input, consisting of both $N$ labeled examples (the *context*) and a new test query. The context is written as $C = \big[(x_1, y_1), (x_2, y_2), \ldots, (x_N, y_N)\big]$, where each pair $(x_i, y_i)$ is drawn from $\mathcal{X} \times \mathcal{Y}$. The test query $x_{\text{test}} \in \mathcal{X}$ is given, and the model predicts its label $y_{\text{test}} \in \mathcal{Y}$ by conditioning on both the context $C$ and $x_{\text{test}}$:

$$y_{\text{test}} \sim P_\theta\Big(y_{\text{test}} \; \Big| \; \underbrace{x_1, y_1, \ldots, x_N, y_N}_{\text{context } C}, x_{\text{test}}\Big). \tag{1}$$

We refer to the mapping that selects the $N$ demonstrations for a given test query as a *sampler* or *retriever*, denoted $\pi\colon \mathcal{X} \to \mathcal{X}^N \times \mathcal{Y}^N$. Designing $\pi$ so that the resulting context maximizes downstream accuracy is central to ICL research and is the focus of this work.

## 2.2 LABEL-FREE CROSS-DOMAIN ICL

In standard settings, samplers are trained and evaluated on data drawn from the same distribution. However, in practical deployments, new tasks and domains appear faster than labeled data can be collected, and privacy or cost constraints often preclude annotating target examples. Retrievers are thus needed to be trained on labeled source domains but operate on unlabeled target domains under distribution shift. Let $\{\mathcal{D}_s\}_{s=1}^S$ be a collection of source-domain distributions on $\mathcal{X} \times \mathcal{Y}$, each yielding labeled pairs $(x, y) \sim \mathcal{D}_s$. A target domain $\mathcal{D}_t$, which is a distribution over $\mathcal{X}$ that provides only unlabeled inputs $x \sim \mathcal{D}_t$.

Formally, recall from the ICL setup that for a test query $x_{\text{test}}$ the sampler $\pi$ selects a context $C = \pi(x_{\text{test}})$ and the model produces a prediction $\hat{y}_{\text{test}}$ according to $P_\theta(y \mid C, x_{\text{test}})$. When applied in the target domain, the objective is to design $\pi$ such that the expected accuracy is maximized,

$$\max_{\pi} \ \mathbb{E}_{x_{\text{test}} \sim \mathcal{D}_t} \Big[ \mathbf{1}\{\hat{y}_{\text{test}} = y_{\text{test}}\} \Big], \tag{2}$$

without access to labels $y$ in $\mathcal{D}_t$ during training. We refer to this problem as *label-free cross-domain ICL*. It highlights the central challenge motivating our analysis: without labels in the target domain, one cannot directly estimate the relevance–diversity–performance relationship, yet this trade-off is essential for effective generalization.

## 3 CROSS-DOMAIN ANALYSIS OF SAMPLER TRADE-OFFS

Most prior samplers have been evaluated in-domain, and even those considering multiple datasets do not provide a systematic analysis of how relevance and diversity correlate with performance across domains. Existing methods typically consider *test-sample relevance*, choosing demonstrations similar to the query (Liu et al., 2021; Rubin et al., 2022; Gupta et al., 2023), or by *in-set diversity*, encouraging dissimilarity among demonstrations (Levy et al., 2022; Ye et al., 2023; Mavromatis et al., 2023; Wang et al., 2024c;b). However, without target-domain labels it is not possible to determine the appropriate trade-off between relevance and diversity or to predict which choice will improve performance. To address this gap, we investigate two guiding questions.

As a starting point for our analysis, we first formalize the two metrics of interest. For a test input $x_{\text{test}}$ with embedding $h(x_{\text{test}})$ and a set of $N$ retrieved demonstrations $C = \{(x_i, y_i)\}_{i=1}^N$, *test-sample relevance* is defined as the average cosine similarity between the test input and the demonstrations:

$$\text{Relevance}(x_{\text{test}}, C) = \frac{1}{N} \sum_{i=1}^{N} \frac{h(x_{\text{test}}) \cdot h(x_i)}{\|h(x_{\text{test}})\| \, \|h(x_i)\|}. \tag{3}$$

*In-set diversity* is defined as the mean pairwise dissimilarity among the demonstration embeddings:

$$\text{Diversity}(C) = \frac{2}{N(N-1)} \sum_{1 \leq i < j \leq N} \left( 1 - \frac{h(x_i) \cdot h(x_j)}{\|h(x_i)\| \, \|h(x_j)\|} \right). \tag{4}$$

Unless otherwise specified, *relevance* and *diversity* in this paper specifically refer to *test-sample relevance* and *in-set diversity* measures.

**Q1:** **Are the relevance–diversity–performance relationships consistent across domains?** To investigate this question, we evaluate several learning-free retrievers, including TOPK-BM25 (Robertson & Zaragoza, 2009), TOPK-BERT (Devlin et al., 2019), and the DPP-BERT method designed for diversity-aware selection (Chen et al., 2018), along with random selection.

We adopt learning-free retrievers in this analysis because they provide an unbiased view of how relevance and diversity relate to performance across domains. Learning-based retrievers (Rubin et al., 2022; Ye et al., 2023; Wang et al., 2024b) often rely on contrastive or task-specific training objectives that bias the embedding space toward a particular relevance–diversity trade-off. For example,

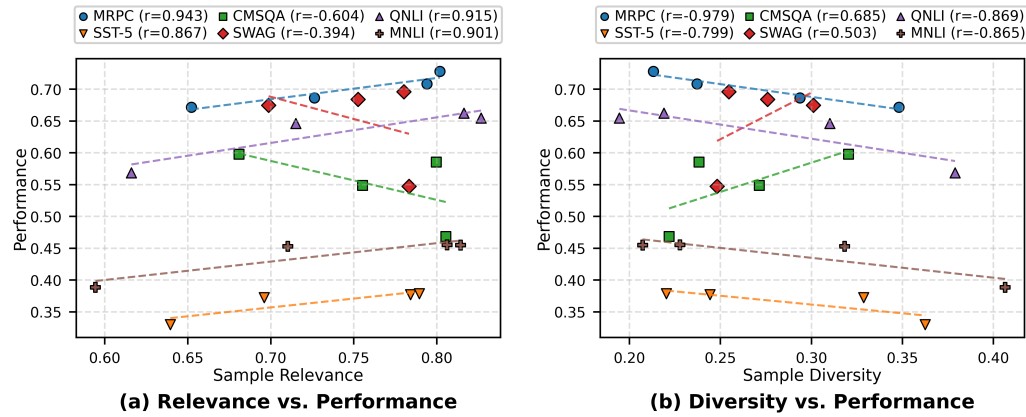

Figure 1: Correlation between sample relevance and performance, and between sample diversity and performance. In the relevance–performance plots, CMSQA and SWAG show negative Pearson values, while the other domains exhibit positive correlations, highlighting that the relationship varies across domains.

embeddings may be explicitly shaped to promote diversity, obscuring the intrinsic correlation between diversity and performance. Learning-free methods, by contrast, allow us to examine these correlations directly without such confounding biases. Using the definitions of relevance and diversity introduced above, we compute the Pearson correlation $r = \frac{\sum_i (x_i - \bar{x})(y_i - \bar{y})}{\sqrt{\sum_i (x_i - \bar{x})^2}\sqrt{\sum_i (y_i - \bar{y})^2}}$ between each metric and ICL accuracy across domains.

As shown in Fig. 1-(a), relevance is strongly and positively correlated with performance in MRPC, SST5, QNLI, and MNLI ($0.86 < r < 0.95$). In contrast, Fig. 1-(b) shows that diversity in the same domains has the opposite effect, with consistently negative correlations ($r < -0.79$). By contrast, CMSQA and SWAG exhibit a reversed pattern: relevance is negatively related to accuracy ($r \approx -0.60$ and $-0.39$), while diversity is positively correlated with performance ($r \approx 0.69$ and $0.50$). Taken together, these results reveal a clear inverse relationship: domains that reward relevance penalize diversity, and those that reward diversity penalize relevance. Thus, relevance and diversity act as opposing signals, and the dominant factor for performance differs by domain.

**A1:** No. The relationships are not consistent across domains. Some datasets favor relevance while others favor diversity, and the two metrics exhibit opposite correlation patterns with performance. This demonstrates that samplers are fundamentally limited in cross-domain generalization. Even when both relevance and diversity are considered, the appropriate trade-off cannot be determined in a label-free setting where the target domain is inaccessible.

**Q2: Do these domain-specific correlations also affect learning-based samplers?** From Q1 we established that the correlations between relevance, diversity, and performance differ by domain, with the two metrics exhibiting opposite trends. A natural question is whether learning-based retrievers, which explicitly optimize these criteria, also inherit such domain-specific patterns. To answer this, we study two representative models: EPR (Rubin et al., 2022) first retrieves a candidate pool of relevant examples using BERT-based embeddings, and then trains a retriever with a contrastive objective. This structure makes relevance the primary criterion and diversity only a secondary adjustment, biasing the model toward relevance-driven retrieval. CEIL (Ye et al., 2023) balances test-sample relevance and in-set diversity using a DPP-based criterion.

Figure 2 summarizes the cross-domain evaluation using relative performance heatmaps. For each training domain, we set the in-domain accuracy to 100, and each entry then indicates the percentage of this baseline that is preserved when the sampler is trained on another domain but evaluated on the same test domain. Thus, rows correspond to training domains and columns to test domains, with the diagonal entries fixed at 100 and the off-diagonal entries capturing cross-domain transfer. This representation makes it possible to directly compare how much performance is lost when moving from in-domain to cross-domain settings. We find that EPR shows the largest gap between

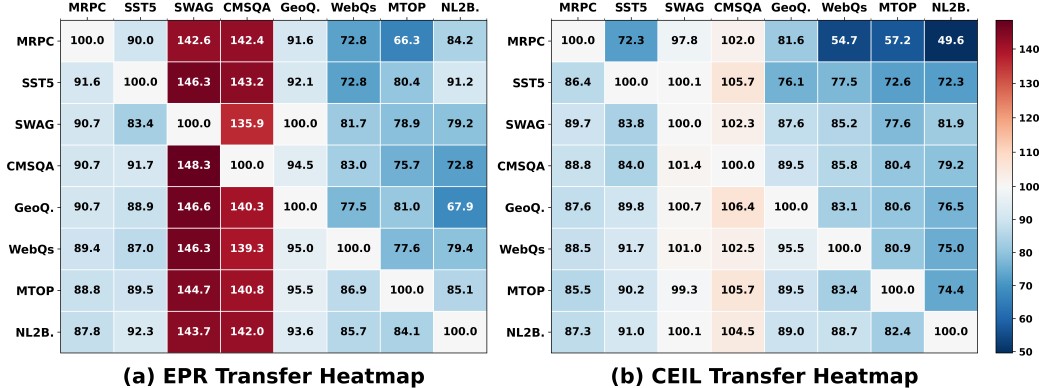

Figure 2: (a) EPR and (b) CEIL sampler performance in cross-domain settings. Each row denotes the source dataset used for training, and each column denotes the target dataset used for evaluation. The performance on the identical source–target pair is normalized to 100%, and the performance on other target domains is expressed as a percentage relative to this baseline.

in-domain and cross-domain performance on diversity-important benchmarks such as SWAG and CMSQA, consistent with the positive diversity–performance correlations in Fig. 1. Because EPR prioritizes relevance during training, its in-domain models fail when relevance is not the dominant signal, and their accuracy can drop even below that of models trained on other domains. CEIL, while explicitly considering both relevance and diversity, exhibits a similar though less extreme tendency: performance still depends heavily on whether the target domain rewards relevance or diversity. An important takeaway is that cross-domain performance is governed by the target domain's relevance–diversity balance, even for learning-based samplers. Cross-domain ICL hinges on how strongly relevance or diversity correlates with performance in the target domain.

**A2:** Yes. Target-specific trade-offs persist in learning-based samplers. EPR prioritizes relevance and fails in diversity-important domains. CEIL incorporates diversity during training and inference yet still shows strong domain dependence. These results indicate that the target domain's relevance–diversity balance is the primary factor of cross-domain ICL.

## 4 LATENT SIGNALS FOR LABEL-FREE CROSS-DOMAIN ADAPTATION

The analyses in Q1 and Q2 establish two key facts. First, relevance and diversity exhibit domain-specific and inverse correlations with performance. Second, cross-domain performance is governed by the interaction between the target domain's relevance–diversity–performance relationship and the weighting of relevance versus diversity in the sampler's design. The central challenge is that, in the absence of target-domain labels, it is unclear how relevance and diversity individually contribute to performance, making it impossible to establish the correct trade-off between them. To address this, we leverage latent representations of LLMs, introducing the *Chain-of-Embedding* (CoE) (Ren et al., 2022; Wang et al., 2024d;e) as a proxy signal.

**Chain-of-Embedding.** Following prior work (Ren et al., 2022; Wang et al., 2024d;e), we interpret an $L$-layer LLM as a sequence of representation transformations. Let the full prompt be denoted by $P = (C, x_{\text{test}})$, consisting of the context $C$ and the test query $x_{\text{test}}$, and suppose $P$ contains $T$ tokens. We denote the hidden state of the $t$-th token at layer $\ell$ by $z_t^{(\ell)}(P) \in \mathbb{R}^d$. A fixed-size sentence-level representation is obtained by mean-pooling across tokens, $h^{(\ell)}(P) = \frac{1}{T} \sum_{t=1}^{T} z_t^{(\ell)}(P)$ for $\ell = 0, \ldots, L$, where $h^{(0)}(P)$ corresponds to the embedding layer and $h^{(L)}(P)$ is the final encoder output. The *Chain-of-Embedding* (CoE) is then defined as the ordered sequence

$$\text{CoE}(P) = \left[ h^{(0)}(P) \to h^{(1)}(P) \to \cdots \to h^{(L)}(P) \right], \tag{5}$$

which can be viewed as a trajectory in $\mathbb{R}^d$ describing how the model's internal representation of the prompt evolves across layers.

**Why CoE is a latent reasoning signal.** Prior work indicates that internal representations of conditional LMs carry rich signals about distributional shift and reasoning quality, which motivates our use of CoE as a proxy for the target domain's retrieval preference. Ren et al. (2022) show that out-of-distribution behavior and selective generation can be detected in conditional language models using model-internal signals beyond surface-form likelihoods, supporting the view that hidden states encode distributional information that is useful without labels. Wang et al. (2024d) demonstrate that the geometry of embedding trajectories during mathematical problem solving is predictive of out-of-distribution status and solution correctness; features derived from the trajectory across layers and steps enable label-free OOD detection. Wang et al. (2024e) formalize the Chain-of-Embedding and use it for output-free self-evaluation, showing that layer-wise hidden-state sequences correlate with answer quality and can score model outputs without reading them. Taken together, these results suggest that CoE behaves as a latent reasoning path: its geometry reflects how the model internally processes an input and therefore can serve as an indirect, label-free signal about task characteristics. In our setting, we leverage this insight to infer whether a target domain rewards relevance or diversity in retrieval.

**Q3: Can CoE geometry provide a label-free indicator of how relevance and diversity affect performance in the target domain?** We analyze CoE trajectories for test prompts from multiple domains and quantify their spread using the PCA hull of top principal components. Specifically, let the combined prompt be denoted by $P = (C, x_{\text{test}})$. For each layer $\ell = 0, \ldots, L$, let $h_\ell(P) \in \mathbb{R}^d$ denote the mean-pooled hidden representation at that layer. Stacking these vectors as rows gives the trajectory matrix $H_P \in \mathbb{R}^{(L+1) \times d}$. To focus on relative movement rather than absolute position, we apply PCA to $H_P$ and project the trajectory onto its top four principal components, obtaining $Z_P \in \mathbb{R}^{(L+1) \times 4}$. The spread of this projected path is quantified by the volume of its PCA hull, denoted $V_{\text{CoE}}(P)$, and domain-level statistics are reported as the mean of $V_{\text{CoE}}(P)$ across queries.

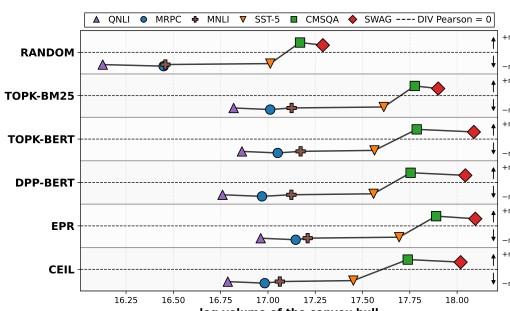

Figure 3: PCA hull volume $V_{\text{CoE}}(P)$ (x-axis) versus the Pearson correlation between diversity and performance (y-axis). Domains with larger $V_{\text{CoE}}(P)$ (e.g., SWAG, CMSQA) favor diversity, while smaller $V_{\text{CoE}}(P)$ (e.g., QNLI, MRPC) favor relevance. The trend is consistent across both learning-free and learning-based retrievers, indicating that $V_{\text{CoE}}(P)$ provides a robust label-free signal of domain characteristics.

Figure 3 plots the PCA hull volume $V_{\text{CoE}}(P)$ (x-axis) against the Pearson correlation between diversity and performance (y-axis). We find that domains where diversity aligns positively with performance (e.g., SWAG, CMSQA) exhibit larger $V_{\text{CoE}}(P)$, indicating broader trajectory exploration. Conversely, domains where relevance dominates (e.g., QNLI, MRPC) show much smaller hull volumes. This relationship holds across different retrievers: among learning-free methods (RANDOM, TOPK-BM25, TOPK-BERT, DPP-BERT) and learning-based samplers (EPR, CEIL), the link between $V_{\text{CoE}}(P)$ and diversity–performance correlation is consistently observed. Thus, the CoE hull volume provides a robust, model-agnostic signal that can be computed without access to a trained sampler.

**A3:** Yes. CoE geometry provides a label-free indicator of whether a target domain rewards relevance or diversity. Domains with broad trajectories tend to favor diversity, while domains with narrow trajectories favor relevance. Crucially, this signal is stable across both learning-free and learning-based retrievers, making it largely independent of model choice. In practice, this means that even without a trained sampler, one can estimate the diversity–performance tendency of a new target domain by measuring $V_{\text{CoE}}(P)$ from simple baselines such as RANDOM or TOPK-BM25 retrieval. Given a pool of existing source datasets, comparing their $V_{\text{CoE}}(P)$ statistics with that of a new target domain allows us to infer the target's trade-off characteristics, enabling label-free adaptation in cross-domain settings. Thus, the Chain-of-Embedding (Wang et al., 2024e) offers a practical and robust signal for guiding sampler adaptation without requiring target-domain labels.

**Interpreting the diversity–performance trade-off for generation in cross-domain settings.** We evaluate cross-domain generation from SST5 to multiple target datasets using the CEIL sampler, a diversity-controllable sampler. We set 0.10 as the base and vary it to probe the trade-off. As shown in Tab. 1, lowering the

Table 1: Effect of the CEIL diversity factor on cross-domain generation performance.

| Source Dataset | SST5 | | | | | | |
|---|---|---|---|---|---|---|---|
| Target Dataset | WEBQS | GEOQUERY | NL2BASH | BREAK | MTOP | SMCALFLOW | AVG |
| 0.01 | 16.63 | 65.36 | 32.23 | 28.63 | 54.00 | 45.22 | 40.35 |
| 0.05 | 14.91 | 61.79 | 28.97 | 27.98 | 53.02 | 44.26 | 38.49 |
| 0.10 | 14.22 | 54.64 | 29.34 | 24.51 | 49.40 | 40.40 | 35.42 |
| 0.5 | 7.09 | 48.93 | 15.26 | 3.81 | 20.04 | 14.51 | 18.27 |
| 1.0 | 9.94 | 44.64 | 20.82 | 12.89 | 37.81 | 28.55 | 25.78 |

diversity factor, and thus prioritizing relevance, consistently improves cross-domain performance across targets. This contrasts with our findings in classification tasks, where larger CoE volume tends to favor diversity through a stronger positive diversity–performance correlation. The results suggest that, under domain shift, generation depends more on format and local context alignment than on broad exploration. In generation, a large CoE volume can reflect style or formatting drift rather than useful semantic exploration, so CoE statistics and the diversity–performance curve do not necessarily align. We therefore recommend prioritizing relevance for generation in cross-domain settings. The CoE volume in the generation task is presented in Appendix D.2.

**Latent Reasoning Path Guidance.** We propose *Latent Reasoning Path Guidance (LRPG)*, which uses the geometry of the Chain-of-Embedding to decide the *direction* of diversity control in a new target domain without accessing labels. See Algorithm 1 for the full procedure in Appendix C.

**Step 0. Construct reference pool.** For each test prompt in source domain $P = (C, x_{\text{test}})$ in $\mathcal{D}_s$, compute $V_{\text{CoE}}(P)$ in the PCA subspace and aggregate over prompts to obtain $\frac{1}{|\mathcal{D}_s|} \sum_{P \in \mathcal{D}_s} V_{\text{CoE}}(P)$, denoted as $v_s$. Let $\mathcal{P}_s$ be a reference pool from source domains, where each entry stores a $v_s$ and the *sign* of its diversity–performance correlation.

**Step 1. Target CoE statistic.** For each test prompt $P = (C, x_{\text{test}})$ in $\mathcal{D}_t$, compute $V_{\text{CoE}}(P)$ in the PCA subspace and aggregate over prompts to obtain $\frac{1}{|\mathcal{D}_t|} \sum_{P \in \mathcal{D}_t} V_{\text{CoE}}(P)$, denoted as $v_t$.

**Step 2. Decide diversity *direction* by matching to $\mathcal{P}_s$.** Order $\mathcal{P}_s$ by $v_s$ as shown in Fig. 3. Let $v_-$ be the largest pool value strictly less than $v_t$ and let $v_+$ be the smallest pool value strictly greater than $v_t$. If the domain at $v_-$ has a *positive* diversity–performance correlation, *increase* diversity in the sampler. If the domain at $v_+$ has a *negative* diversity–performance correlation, *decrease* diversity in the sampler. If both conditions hold, choose the smaller step toward the nearer entry in $\mathcal{P}_s$.

For generation tasks, default to emphasizing relevance by decreasing diversity unless the target statistic clearly aligns with a diversity-favoring bracket. This procedure selects a direction without target labels. It leaves the sampler's scoring form unchanged and provides a high-level rule for whether to encourage or suppress diversity when transferring to a new domain.

## 5 RELATED WORK

Large language models (LLMs) exhibit the emergent ability of *in-context learning* (ICL), where a model solves new tasks by conditioning on a few labeled examples without updating parameters (Brown et al., 2020; Wei et al., 2022). A number of surveys provide a comprehensive overview of ICL and its applications (Dong et al., 2022). In this work we focus on the demonstration selection problem, often referred to as the *sampler* or retriever, which has been shown to critically affect downstream performance (Liu et al., 2021). Early work studied simple heuristics that operate without additional training. Liu et al. (2021) highlight the importance of example choice, while Sorensen et al. (2022) use mutual information as a label-free criterion. Coverage (Gupta et al., 2023), skill relevance (An et al., 2023), and entropy-based trategies (Peng et al., 2024) have also been explored. Other methods leverage iterative demonstration selection (Qin et al., 2023) or subset-level selection for prompt generation (Purohit et al., 2025). Such approaches are attractive due to their simplicity but lack adaptability across domains. To overcome these limitations, learning-based retrievers optimize trainable metrics for demonstration selection. EPR (Rubin et al., 2022) learns to retrieve relevant prompts, while CEIL (Ye et al., 2023) models the joint probability of example sets using determinantal point processes (DPP) (Chen et al., 2018). Extensions include mixture-of-demonstrations (Wang et al., 2024b), unified retrievers trained across multiple tasks (Li et al., 2023), sequential example selection (Liu et al., 2024a), and RL-based demonstration retrievers (Zhang et al., 2022; Scarlatos & Lan, 2023; Wang et al., 2024c). Additional works incorporate uncertainty and diversity trade-

Table 2: Label-free cross-domain transfer to diverse target datasets. The retriever is trained on SST5 and evaluated on diverse targets without access to target labels. The table highlights CEIL+LRPG, our CoE-based rule that decides whether to increase or suppress diversity for each target, and CEIL+LRPG-REV, which applies the opposite decision. It also reports gains relative to CEIL.

| Method | Classification | | | | | Generation | | | | | | Avg. |
|---|---|---|---|---|---|---|---|---|---|---|---|---|
| | MRPC | QNLI | MNLI | CMSQA | SWAG | WebQs | GeoQ. | NL2Bash | Break | MTOP | SMCal. | |
| *Learning-free* | | | | | | | | | | | | |
| RANDOM | 67.16 | 56.84 | 38.85 | **59.79** | 67.46 | 4.92 | 33.93 | 14.81 | 1.83 | 6.67 | 3.84 | 32.37 |
| TOPK-BM25 | 68.63 | 64.62 | 45.30 | 54.87 | 68.37 | 16.63 | 62.86 | 34.48 | 25.41 | **54.09** | 44.42 | 49.06 |
| TOPK-BERT | **72.79** | 65.46 | 45.49 | 46.85 | 54.71 | **17.18** | **66.79** | 31.36 | 26.71 | 53.47 | **45.52** | 47.85 |
| DPP-BERT | 70.83 | 66.21 | 45.54 | 58.56 | **69.60** | 15.11 | 64.29 | **35.83** | 26.78 | 53.42 | 45.50 | 50.15 |
| *Learning-based* | | | | | | | | | | | | |
| EPR | 71.32 | 62.07 | 45.73 | 59.38 | 66.73 | 14.52 | 66.07 | 30.44 | 23.97 | 42.64 | 38.93 | 47.44 |
| CEIL | 69.85 | 62.38 | 45.77 | 59.30 | 69.20 | 14.22 | 54.64 | 29.34 | 24.51 | 49.40 | 40.40 | 47.18 |
| CEIL+LRPG | 68.87 | **68.63** | **46.80** | 59.79 | 68.20 | 16.63 | 65.36 | 32.23 | **28.63** | 54.00 | 45.22 | **50.40** |
| CEIL+LRPG-REV | 64.95 | 55.39 | 42.60 | 59.62 | 68.53 | 9.94 | 44.64 | 20.82 | 12.89 | 37.81 | 28.55 | 40.52 |
| GAIN OVER CEIL | −0.98 | +6.25 | +1.03 | +0.49 | −1.00 | +2.41 | +10.72 | +2.89 | +4.12 | +4.60 | +4.82 | +3.22 |
| GAIN OF REV OVER CEIL | −4.90 | −6.99 | −3.17 | +0.32 | −0.67 | −4.28 | −10.00 | −8.52 | −11.62 | −11.59 | −11.85 | −6.66 |

offs (Mavromatis et al., 2023; Yang et al., 2023; Levy et al., 2022). These methods often achieve strong in-domain performance but their generalization to unseen domains remains underexplored.

# 6 EXPERIMENTS

**Implementation Details.** We use GPT-Neo (Black et al., 2022), a large language model with 2.7 billion parameters pre-trained on the Pile (Gao et al., 2020), an 825 GB diverse text corpus collected from various high-quality sources.

**Baselines.** Our baselines include both learning-free and learning-based retrievers. (1) **RANDOM** randomly selects in-context examples from the training set without repetition. (2) **TOPK-BM25** (Robertson & Zaragoza, 2009) uses BM25, based on TF-IDF, to retrieve the Top-$K$ highest-scoring examples for the test query. (3) **TOPK-BERT** (Devlin et al., 2019) is a dense retriever based on BERT (Wolf et al., 2020). (4) **DPP-BERT** (Chen et al., 2018) performs exact greedy MAP inference for Determinantal Point Processes with incremental Cholesky updates, reducing complexity from $\mathcal{O}(M^4)$ to $\mathcal{O}(M^3)$ (where $M$ is the number of candidates) and $\mathcal{O}(N^2 M)$ to select $N$ examples, using a sliding-window strategy for relevance–diversity balance. (5) **EPR** (Rubin et al., 2022) trains a contrastive dense retriever to select relevant and informative prompts by encoding queries and candidates into a shared embedding space. (6) **CEIL** (Ye et al., 2023) jointly selects examples via a conditional DPP trained with a contrastive loss to align subset scores to language model preferences, using efficient MAP inference.

To verify our method, we use the diversity scale factors from DPP-BERT and CEIL. Since we cannot directly quantify how much to increase or decrease diversity, we adopt a simple rule: scale the DPP diversity–relevance trade-off parameter by $\times 10$ to encourage diversity or by $\times 0.1$ to suppress it. Despite its simplicity, this adjustment is effective in practice and yields consistent improvements in our experiments. For details on how DPP-BERT and CEIL adjust diversity, see Appendix C.

**Label-free cross-domain adaptation with LRPG.** Table. 2 evaluates a CEIL retriever trained on SST-5 and adapted to diverse targets using LRPG, which measures each target's CoE geometry and decides whether to increase or suppress diversity during sampling without accessing target labels. CEIL+LRPG consistently improves over CEIL and achieves the best overall average (+3.22), with notable gains on representative targets such as QNLI, MTOP, SMCalFlow, and Geo-Query. To test directionality, we also apply a *reversed* policy that flips the increase/suppress decision (CEIL+LRPG-REV). The separate gain row shows broad degradations relative to CEIL (e.g., Break −11.62, SMCalFlow −11.85, MTOP −11.59, GeoQuery −10.00), with only a negligible improvement on CMSQA (+0.32). This contrast underscores that the CoE-guided sign of diversity adjustment matters and that the proposed rule aligns with target-domain retrieval preferences under label-free cross-domain transfer. Additional cross-domain experiments of our approach are provided in Appendix E.

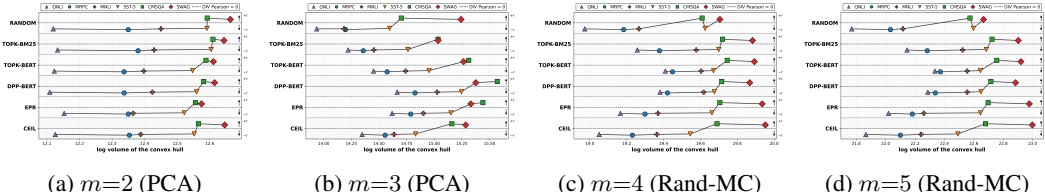

(a) $m{=}2$ (PCA)     (b) $m{=}3$ (PCA)     (c) $m{=}4$ (Rand-MC)     (d) $m{=}5$ (Rand-MC)

Figure 4: **Robustness across projection choices.** PCA projections at lower dimensions $m{=}2$ and $m{=}3$ retain the qualitative ordering between hull log volume and the diversity–performance correlation observed at $m{=}4$. Rand-MC averaging over 100 random orthonormal projections preserves the same ordering at $m{=}4$ and $m{=}5$.

**Diversity scale ablation on a learning-free sampler.** We study how performance varies with the diversity parameter in DPP-BERT and how this trend relates to CoE geometry. We evaluate MRPC and SWAG for classification and WebQS and GeoQuery for generation. As shown in Tab. 3, SWAG, which has a large CoE hull volume, benefits from higher diversity, whereas MRPC improves as diversity is reduced. Generation tasks such as WebQS and GeoQuery consistently favor relevance, exhibiting a negative diversity–performance correlation. Taken together, these findings indicate that our analysis generalizes to both learning-based and learning-free retrieval methods.

Table 3: DPP-BERT diversity scale sweep on a subset of targets.

| Method | MRPC | SWAG | WebQs | GeoQuery |
|---|---|---|---|---|
| 0.0025 | 70.83 | 68.96 | 17.27 | 69.29 |
| 0.0050 | 70.34 | 69.13 | 17.47 | 67.14 |
| 0.0100 | 71.81 | 69.32 | 16.24 | 65.00 |
| 0.0250 | 72.06 | 69.54 | 15.99 | 64.64 |
| 0.0500 | 70.83 | 69.60 | 15.11 | 64.29 |
| 0.1000 | 69.85 | 69.66 | 14.57 | 57.14 |
| 0.2500 | 69.36 | 69.77 | 13.93 | 52.50 |
| 0.5000 | 69.85 | 69.77 | 13.83 | 48.93 |
| 1.0000 | 70.34 | 69.76 | 13.88 | 49.64 |
| Div-Perf Corr | -0.979 | 0.503 | -0.937 | -0.927 |

**Robustness of CoE hull volume to projection choice** The $m$-dimensional volume of the CoE trajectory serves as our measure for predicting the diversity–performance correlation. Summarizing CoE spread with a PCA hull is intuitive because an LLM maps input tokens into a sequence of hidden states and then returns to the token space to generate outputs, which yields loop-like excursions in low-dimensional views. These loops are well captured by the PCA hull envelope, as in Fig. 5, and this perspective is consistent with prior work (Ren et al., 2022; Wang et al., 2024d;e).

We then examine robustness of this measure to the projection choice. (1) *PCA dimensional robustness.* Using low-dimensional subspaces with $m{=}2$ or $m{=}3$ yields the same qualitative dataset ordering and the same diversity–performance relations as the $m{=}4$ analysis, see Fig. 4. (2) *Random projection robustness.* Replacing principal axes with random orthonormal directions yields the same pattern. For each prompt we draw $R \in \mathbb{R}^{d \times m}$ with i.i.d. $\mathcal{N}(0, 1/m)$ entries, form $Z = H_P R$, compute the hull log $m$-volume, and average over 100 projections with $m \in \{4, 5\}$. The ordering and correlations are maintained, as summarized in Fig. 4. Across both methods and these dimensions, the alignment with Fig. 3 indicates that the CoE spread can be measured robustly.

## 7 CONCLUSION

We studied the problem of label-free cross-domain in-context learning, where a sampler trained on source domains must operate in target domains without labels. Our analysis showed that the relevance–diversity–performance relationship is highly domain dependent and limits the generalization of existing samplers. We further demonstrated that the geometry of the Chain-of-Embedding (CoE) offers a robust, model-agnostic signal of whether a domain favors relevance or diversity. Building on this insight, we proposed *Latent Reasoning Path Guidance (LRPG)*, which leverages CoE statistics to guide samplers in adjusting their diversity component. Experiments across diverse benchmarks confirmed that LRPG alleviates the severe performance drops of existing methods and consistently improves ICL accuracy in cross-domain settings. In addition, we provide a simple CoE-based criterion that predicts when diversity helps, validate it across models and domains, and show that LRPG is plug-and-play with existing samplers with negligible overhead. Together, these results show that CoE geometry serves as a practical control signal for label-free adaptation and that LRPG offers an orthogonal enhancement for both learning-free and learning-based samplers.

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

## A  EXPERIMENTAL SETUP

### A.1  BASELINES

We consider both learning-free and learning-based retrievers as baselines. We illustrate the detail of each baseline as follows.

**RANDOM**  : A retriever that randomly draws in-context examples from the training set, with no repetition.

**TOPK-BM25**  : BM25 (Robertson & Zaragoza, 2009) is a retrieval method that extends TF-IDF. We take the Top-K highest-scoring examples as the in-context set.

**TOPK-BERT**  : A dense retriever that relies on BERT embeddings (Devlin et al., 2019). We use `bert-base-uncased`[1], which is publicly available in Hugging Face Transformers (Wolf et al., 2020).

**DPP-BERT**  : A DPP-based retriever that reuses the original BERT embeddings without fine-tuning, and applies MAP inference to select a subset (Chen et al., 2018).

**EPR**  : A learning-based dense retriever optimized to return a high-quality in-context example (Rubin et al., 2022). At inference time, it chooses the Top-K most similar examples and extend the approach to tasks beyond semantic parsing following Rubin et al. (2022).

**CEIL**  : A learning-based retriever that formulates in-context example selection as a subset selection problem. It instantiates a determinantal point process (DPP) to jointly capture input–exemplar relevance and inter-exemplar diversity, and it trains the DPP kernel with a contrastive objective that incorporates LM preference signals (Ye et al., 2023).

### A.2  DATASETS

We evaluate on twelve tasks spanning *classification* and *generation*, following the setup of Ye et al. (2023). Below we summarize each dataset by task type.

#### A.2.1  CLASSIFICATION

**SST-5** (Socher et al., 2013) is a five-way sentiment benchmark with labels *very positive*, *positive*, *neutral*, *negative*, and *very negative*.

**MRPC** (Dolan et al., 2004) contains sentence pairs from news sources with human judgments of semantic equivalence. The goal is to predict whether the two sentences convey the same meaning.

**MNLI** (Williams et al., 2018) is a large natural language inference corpus. Given a premise and a hypothesis, the model predicts *entailment*, *contradiction*, or *neutral*.

**QNLI** (Wang et al., 2018) reframes question answering as sentence selection. Given a question and a context sentence, the task is to decide if the sentence contains the answer.

**CMSQA** (Talmor et al., 2019) is a five-choice multiple-choice benchmark that requires diverse commonsense knowledge to select the correct answer.

**SWAG** (Zellers et al., 2018) is a commonsense reasoning benchmark with four answer choices and adversarial distractors. The distractors are drawn from captioning corpora such as ActivityNet Captions (Heilbron et al., 2015) and LSMDC (Rohrbach et al., 2017), which makes the options particularly challenging to distinguish.

#### A.2.2  GENERATION

**WebQuestions (WebQs)** (Berant et al., 2013) contains questions sourced from the web whose answers are entities in Freebase. The model must produce the correct entity as the output.

---

[1]https://huggingface.co/bert-base-uncased

Table 4: Summary of datasets, tasks, and experimental settings. #ICE denotes the mean number of in-context examples per validation instance.

| Type | Dataset | Task | #Train | #Validation | #ICE |
|---|---|---|---|---|---|
| Classification | SST-5 (Socher et al., 2013) | Sentiment Analysis | 8,534 | 1,101 | 40 |
| | MRPC (Dolan et al., 2004) | Paraphrase Detection | 3,668 | 408 | 27 |
| | MNLI (Williams et al., 2018) | Natural Language Inference | 392,568 | 19,647 | 40 |
| | QNLI (Wang et al., 2018) | Natural Language Inference | 104,707 | 5,463 | 27 |
| | CMSQA (Talmor et al., 2019) | Commonsense Reasoning | 9,740 | 1,221 | 50 |
| | SWAG (Zellers et al., 2018) | Commonsense Reasoning | 52,611 | 20,006 | 50 |
| Generation | WebQs (Berant et al., 2013) | Open-Domain QA | 3,778 | 2,032 | 50 |
| | GeoQuery (Zelle & Mooney, 1996) | Code Generation | 404 | 280 | 50 |
| | NL2Bash (Lin et al., 2018) | Code Generation | 7,441 | 609 | 43 |
| | Break (Wolfson et al., 2020) | Semantic Parsing | 44,184 | 7,760 | 28 |
| | MTOP (Li et al., 2021) | Semantic Parsing | 15,564 | 2,235 | 41 |
| | SMCalFlow (Andreas et al., 2020) | Semantic Parsing | 102,491 | 14,751 | 22 |

**NL2Bash** (Lin et al., 2018) pairs natural language descriptions with Bash commands. Each example consists of an expert-written description and a target command collected from the web, and the task is to generate the command from the description.

**GeoQuery** (Zelle & Mooney, 1996; Shaw et al., 2021) maps English questions about US geography to executable Prolog queries. It uses the original `Standard` split and the compositional splits introduced by Shaw et al. (2021) to test compositional generalization: `Template` where abstract output templates do not overlap between train and test (Finegan-Dollak et al., 2018), `TMCD` which maximizes distribution shift in compound structures, and `Length` where test instances are longer than those in training.

**Break** (Wolfson et al., 2020) transforms complex questions into a sequence of atomic operations that form a language-based meaning representation. Following Rubin et al. (2022), we use the low-level Break subset.

**MTOP** (Li et al., 2021) is a multilingual task-oriented semantic parsing benchmark spanning six languages and eleven domains. Outputs are complex queries with nested intent–slot structures. In line with prior work (Rubin et al., 2022), we report results on the English subset.

**SMCalFlow** (Andreas et al., 2020) convert natural utterances about calendars, weather, locations, and people into executable dataflow programs with API calls, function composition, and constraints. SMCalFlow-CS citepyin2021compositional focuses on single-turn inputs across two domains—organization structure and event creation—with disjoint symbol sets. We evaluate on the cross-domain $C$ and single-domain $S$ test sets. In few-shot settings with split $k$-C where $k \in \{8, 16, 32\}$, the training set includes $k$ additional cross-domain examples that expose the necessary composition symbols.

## B  ADDITIONAL BACKGROUND AND RELATED WORK

**Analyses of ICL.** Several studies have examined the mechanisms of ICL itself. Work on prompt sensitivity (Lu et al., 2021), probing decision boundaries (Zhao et al., 2024), and identifying the layers responsible for ICL (Sia et al., 2024; Cho et al., 2024) provide insights into why demonstrations affect model behavior. Other analyses study the role of demonstrations (Min et al., 2022; Olsson et al., 2022), implicit Bayesian inference views (Xie et al., 2021), and how pretrained representations contribute to downstream ICL (Wei et al., 2021). In contrast to these works, which primarily probe where and how ICL emerges inside the model, our analysis targets the cross-domain behavior of samplers: we quantify domain-specific, approximately linear correlations between relevance/diversity and performance, show that these trade-offs are target-dependent even for learning-based samplers, and use hidden-state trajectory as a *label-free, model-agnostic* indicator of which side of the trade-off is preferred in a new domain. Thus, our focus is not on explaining ICL mechanisms, but on diagnosing and adapting the *retrieval policy* that governs demonstration selection under distribution shift.

**Meta ICL.** Meta-ICL methods (Min et al., 2021; Coda-Forno et al., 2023; Li et al., 2024b; Goddard et al., 2025; Zhuang et al., 2024; Wang et al., 2024a; Wu et al., 2025; Zhang et al., 2025; Li et al., 2024a; Xu et al., 2024) and unsupervised domain adaptation retrieval approaches (Long et al., 2023; Liu et al., 2024b; Vettoruzzo et al., 2024) typically assume training time access to source data. They leverage unlabeled target inputs to augment prompts or to define auxiliary objectives that adapt the sampler. Our setting is stricter. The sampler operates in an unlabeled target domain with no training access and it relies only on hidden state Chain of Embedding trajectories at inference. This difference limits direct comparisons. We do not require a new meta retriever and we do not modify the language model. We derive a guidance signal from CoE geometry that informs the relevance–diversity balance of a sampler at test time. The signal can be used in a training free mode at test time and it also admits a training based variant.

## C  IMPLEMENTATION DETAILS

To verify that our method works effectively across different samplers, we conducted experiments on both learning-free and learning-based samplers, focusing on DPP-BERT and CEIL, which allow diversity adjustment. The DPP-BERT sampler computes a relevance score matrix between the test input embedding and demonstration embeddings extracted from the `bert-base-uncased` model, and normalizes this matrix with a scale factor to account for diversity. The CEIL sampler employs a pretrained EPR sampler to select demonstration sets using the DPP-BERT strategy. It then compares the generated outputs with ground-truth labels and assigns accuracy-based scores to each demonstration set. The sampler is trained using an alignment loss between the DPP score and the assigned scores of the demonstration sets. During training, when computing the relevance score matrix, CEIL normalizes it with a scale factor in the same manner as DPP-BERT to incorporate diversity. At inference, CEIL also applies the DPP-BERT algorithm to perform diversity-aware sampling. In summary, DPP-BERT utilizes the diversity scale factor only during inference, whereas CEIL leverages it in both training and inference.

**Determinantal Point Processes for ICL.** Determinantal point processes (DPPs) model subset selection with repulsion, which naturally captures diversity (Kulesza et al., 2012). Given item features $\{\mathbf{a}_i\}_{i=1}^M$ and a positive semi–definite kernel $\mathbf{L}$ with $\mathbf{L}_{ij} = k(\mathbf{a}_i, \mathbf{a}_j)$, the probability of a subset $S \subseteq \{1, \ldots, M\}$ is

$$\mathcal{P}(S) = \frac{\det(\mathbf{L}_S)}{\det(\mathbf{L} + \mathbf{I})}.$$

To inject both *relevance* to a test input $\mathbf{x}$ and *diversity* among items, a conditional kernel is used (Ye et al., 2023):

$$\tilde{k}(\mathbf{a}_i, \mathbf{a}_j \mid \mathbf{x}) = g(\mathbf{a}_i, \mathbf{x})\, k(\mathbf{a}_i, \mathbf{a}_j)\, g(\mathbf{a}_j, \mathbf{x}), \quad \tilde{\mathbf{L}} = \mathrm{Diag}(\mathbf{r})\, \mathbf{L}\, \mathrm{Diag}(\mathbf{r}),$$

where $r_i = g(\mathbf{a}_i, \mathbf{x})$ is the relevance score. The unnormalized log score for subset $S$ decomposes as

$$\log \det(\tilde{\mathbf{L}}_S) = \sum_{i \in S} \log r_i^2 + \log \det(\mathbf{L}_S),$$

which makes the relevance–diversity contributions explicit. A trade-off is introduced by

$$\log \det(\mathbf{L}_S') = \frac{1}{\lambda} \sum_{i \in S} r_i + \log \det(\mathbf{L}_S), \quad \mathbf{L}' = \mathrm{Diag}\big( \exp(\tfrac{\mathbf{r}}{2\lambda}) \big)\, \mathbf{L}\, \mathrm{Diag}\big( \exp(\tfrac{\mathbf{r}}{2\lambda}) \big),$$

so that $\lambda$ controls the relative weight on relevance. In practice, two encoders (for $\mathbf{x}$ and $\mathbf{a}$) produce embeddings and dot products serve as $g$ and $k$.

**Algorithm LRPG.** Algorithm 1 summarizes LRPG. From labeled source domains we build a reference pool $\mathcal{P}_s$ by computing the CoE hull volume for each prompt, taking the mean per domain, and recording the sign of the diversity–performance correlation. We sort $\mathcal{P}_s$ by the mean volume. For an unlabeled target domain, we compute its mean CoE hull volume in an $m$-dimensional PCA subspace and locate the nearest smaller and larger entries in $\mathcal{P}_s$. If the nearest smaller entry favors diversity we choose INCREASE. If the nearest larger entry penalizes diversity we choose DECREASE. For generation tasks we default to DECREASE. Finally, we adjust the sampler's diversity scale multiplicatively, setting $s_{\mathrm{new}} = 10 \times s_{\mathrm{base}}$ for INCREASE or $s_{\mathrm{new}} = 0.1 \times s_{\mathrm{base}}$ for DECREASE.

---

**Algorithm 1** Latent Reasoning Path Guidance (LRPG)

---

**Require:** Labeled source domains $\{\mathcal{D}_s\}_{s=1}^S$, unlabeled target domain $\mathcal{D}_t$, PCA projection dimension $m$, task flag $\texttt{isGen} \in \{\texttt{true}, \texttt{false}\}$, baseline diversity scale $s_{\text{base}}$
**Ensure:** Diversity direction $\text{DIR} \in \{\text{INCREASE}, \text{DECREASE}\}$ and applied scale $s_{\text{new}}$
    **Step 0. Construct reference pool**
1: $\mathcal{P}_s \leftarrow \emptyset$
2: **for** each source domain $\mathcal{D}_s$ **do**
3:      $v_s \leftarrow \frac{1}{|\mathcal{D}_s|} \sum_{P \in \mathcal{D}_s} V_{\text{CoE}}(P)\}$
4:      $\sigma_s \leftarrow \text{sign}\big(\text{corr}(\text{diversity}, \text{performance})\big)$ with $\sigma_s \in \{+1, -1\}$
5:      add $(v_s, \sigma_s)$ to $\mathcal{P}_s$
6: **end for**
7:
    **Step 1. Target CoE statistic**
8: sort $\mathcal{P}_s$ by $v_s$ in ascending order
9: $v_t \leftarrow \frac{1}{|\mathcal{D}_t|} \sum_{P \in \mathcal{D}_t} V_{\text{CoE}}(P)$
10:
    **Step 2. Decide diversity *direction* by matching to $\mathcal{P}_s$**
11: $(v_-, \sigma_-) \leftarrow \underset{(v_i, \sigma_i) \in \mathcal{P}_s \,:\, v_i < v_t}{\arg \max} v_i$
12: $(v_+, \sigma_+) \leftarrow \underset{(v_i, \sigma_i) \in \mathcal{P}_s \,:\, v_i > v_t}{\arg \min} v_i$
13: **if** $\sigma_- = +1$ **then**
14:      $\text{DIR} \leftarrow \text{INCREASE}$
15: **else if** $\sigma_+ = -1$ **then**
16:      $\text{DIR} \leftarrow \text{DECREASE}$
17: **else**
18:      let $(v_{\text{nn}}, \sigma_{\text{nn}})$ be the nearer neighbor in $\mathcal{P}_s$ to $v_t$
19:      $\text{DIR} \leftarrow \text{INCREASE}$ if $\sigma_{\text{nn}} = +1$ else $\text{DIR} \leftarrow \text{DECREASE}$
20: **end if**
21: **if** $\texttt{isGen} = \texttt{true}$ **then**
22:      $\text{DIR} \leftarrow \text{DECREASE}$
23: **end if**
24: $s_{\text{new}} \leftarrow s_{\text{base}} \times 10$ if $\text{DIR} = \text{INCREASE}$ else $s_{\text{base}} \times 0.1$
25: **return** $s_{\text{new}}$

---

# D ANALYSIS DETAILS

## D.1 Q1: FULL QUANTITATIVE RESULTS

Table 5 reports the quantitative results for Q1 in the Analysis section. We measure the relationships between sample relevance and performance, and between sample diversity and performance, across six classification datasets using learning-free methods. For readability, the relevance and diversity values are each listed in ascending order.

Table 5: Quantitative results corresponding to the Analysis Q1 graph: (a) relevance vs. performance and (b) diversity vs. performance for six datasets using learning-free samplers.

(a) Relevance → Performance

| Dataset | Method | Relevance | Performance |
|---|---|---|---|
| MRPC | RANDOM | 0.6522 | 67.16 |
| | TOPK-BM25 | 0.7263 | 68.63 |
| | DPP-BERT | 0.7941 | 70.83 |
| | TOPK-BERT | 0.8020 | 72.79 |
| SST-5 | RANDOM | 0.6395 | 32.97 |
| | TOPK-BM25 | 0.6961 | 37.24 |
| | DPP-BERT | 0.7843 | 37.69 |
| | TOPK-BERT | 0.7896 | 37.87 |
| CMSQA | RANDOM | 0.6808 | 59.79 |
| | TOPK-BM25 | 0.7553 | 54.87 |
| | DPP-BERT | 0.7997 | 58.56 |
| | TOPK-BERT | 0.8055 | 46.85 |
| SWAG | RANDOM | 0.6988 | 67.46 |
| | TOPK-BM25 | 0.7527 | 68.37 |
| | DPP-BERT | 0.7803 | 69.60 |
| | TOPK-BERT | 0.7834 | 54.71 |
| QNLI | RANDOM | 0.6161 | 56.84 |
| | TOPK-BM25 | 0.7151 | 64.62 |
| | DPP-BERT | 0.8164 | 66.21 |
| | TOPK-BERT | 0.8268 | 65.46 |
| MNLI | RANDOM | 0.5944 | 38.85 |
| | TOPK-BM25 | 0.7103 | 45.30 |
| | DPP-BERT | 0.8063 | 45.54 |
| | TOPK-BERT | 0.8144 | 45.49 |

(b) Diversity → Performance

| Dataset | Method | Diversity | Performance |
|---|---|---|---|
| MRPC | TOPK-BERT | 0.2132 | 72.79 |
| | DPP-BERT | 0.2372 | 70.83 |
| | TOPK-BM25 | 0.2937 | 68.63 |
| | RANDOM | 0.3481 | 67.16 |
| SST-5 | TOPK-BERT | 0.2203 | 37.87 |
| | DPP-BERT | 0.2443 | 37.69 |
| | TOPK-BM25 | 0.3287 | 37.24 |
| | RANDOM | 0.3626 | 32.97 |
| CMSQA | TOPK-BERT | 0.2218 | 46.85 |
| | DPP-BERT | 0.2382 | 58.56 |
| | TOPK-BM25 | 0.2713 | 54.87 |
| | RANDOM | 0.3204 | 59.79 |
| SWAG | TOPK-BERT | 0.2482 | 54.71 |
| | DPP-BERT | 0.2547 | 69.60 |
| | TOPK-BM25 | 0.2759 | 68.37 |
| | RANDOM | 0.3011 | 67.46 |
| QNLI | TOPK-BERT | 0.1945 | 65.46 |
| | DPP-BERT | 0.2190 | 66.21 |
| | TOPK-BM25 | 0.3101 | 64.62 |
| | RANDOM | 0.3790 | 56.84 |
| MNLI | TOPK-BERT | 0.2072 | 45.49 |
| | DPP-BERT | 0.2277 | 45.54 |
| | TOPK-BM25 | 0.3182 | 45.30 |
| | RANDOM | 0.4065 | 38.85 |

## D.2 Q3: FULL QUANTITATIVE RESULTS, PROJECTION ROBUSTNESS, AND COMPUTATION

**Overview.** We summarize how broadly a prompt explores representation space along its Chain-of-Embedding (CoE) using a PCA hull measure computed after projecting the centered trajectory to a low-dimensional subspace obtained by PCA. This measure captures the envelope of reachable states for the prompt. Empirically, the same dataset ordering emerges across a range of projection choices, indicating that the signal is not tied to a particular basis selection.

**Quantitative results.** The full quantitative results for Q3 in the Analysis section are shown in Tabs. 6 to 11. For both learning-free and learning-based methods, we project onto a 4D subspace using the top singular directions and compute the PCA hull *log volume* of the projected CoE trajectory as our reasoning-path measure. Dataset-wise diversity–performance Pearson $r$ values are taken from Q1. We also visualize hidden-state trajectories using PCA. Using 100 prompts selected by the RANDOM method, we project to 2D via the top two singular directions and show the results in Fig. 5. CMSQA and SWAG explore a broader region of the representation space, whereas datasets with a strongly negative diversity–performance correlation (e.g., MRPC) concentrate in a smaller area. While the main text uses projection dimension $m=4$, Fig. 6 shows that reducing the

projection to $m{=}2$ or $m{=}3$ still provides a meaningful discrimination signal with respect to the diversity–performance Pearson correlation, yielding the same qualitative ordering across datasets and retrievers.

**Projection robustness via Rand-MC.** To confirm that the effect is not an artifact of a particular axis choice, we also examine the geometry of hidden states using a Random-Projection Monte-Carlo (Rand-MC) procedure. For each CoE trajectory, we perform 100 independent Gaussian random projections, compute the convex hull *log $m$-volume* in each projected space, and average the results. Fig. 7 with projection dimensions $m{=}4, 5, 6$ corroborates the PCA-based findings. Across retrievers and datasets, the dataset ordering remains consistent under random orthonormal bases. This indicates that the convex hull log volume offers a practical, label-free summary of how broadly a prompt explores the representation space while being reasonably stable across projection choices.

---

**Algorithm 2** Rand-MC estimator of CoE hull log $m$-volume

---

**Require:** CoE trajectory $H_P \in \mathbb{R}^{(L+1) \times d}$, where $d$ is the hidden-state dimension and $L$ is the number of transformer layers (the trajectory has $L{+}1$ points including the embedding layer). Projection dimension $m$ with $1 \leq m < \min\{d, L{+}1\}$. Number of random projections $n_{\text{proj}} \in \mathbb{N}$.
**Ensure:** $\widehat{V}_{\text{CoE}}^{(m)}(P)$, the average convex hull log $m$-volume
1: **for** $j = 1$ to $n_{\text{proj}}$ **do**
2:    Sample $R \in \mathbb{R}^{d \times m}$ with i.i.d. $\mathcal{N}(0, 1/m)$ entries
3:    Project the trajectory: $Z \leftarrow H_P R \in \mathbb{R}^{(L+1) \times m}$
4:    Compute $v_j \leftarrow \text{Vol}_m\big(\text{Hull}(Z)\big)$ using a convex hull routine such as Quickhull
5:    Record $s_j \leftarrow \log v_j$
6: **end for**
7: **return** $\widehat{V}_{\text{CoE}}^{(m)}(P) \leftarrow \frac{1}{n_{\text{proj}}} \sum_{j=1}^{n_{\text{proj}}} s_j$

---

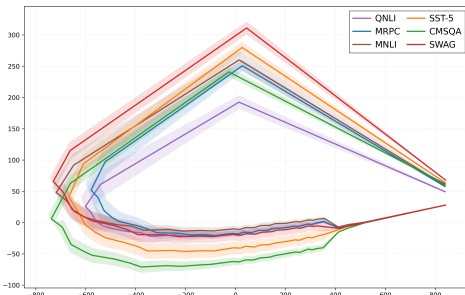

Figure 5: **CoE envelope in 2D.** Projecting to $m{=}2$ for 100 prompts sampled by RANDOM reveals loop-like envelopes in low-dimensional views. Broader envelopes (e.g., SWAG, CMSQA) align with stronger diversity–performance correlations, while MRPC concentrates in a smaller region.

Table 6: PCA hull volume in 2 dimension (higher means a larger hull). The bottom row reports the dataset-wise diversity–performance correlation ($r$).

| | **DATASET** | | | | | |
|---|---|---|---|---|---|---|
| **METHOD** | QNLI | MRPC | MNLI | SST-5 | CMSQA | SWAG |
| RANDOM | 12.12 | 12.35 | 12.45 | 12.59 | 12.59 | 12.66 |
| TOPK-BM25 | 12.13 | 12.38 | 12.43 | 12.60 | 12.61 | 12.64 |
| TOPK-BERT | 12.12 | 12.34 | 12.40 | 12.55 | 12.59 | 12.61 |
| DPP-BERT | 12.11 | 12.34 | 12.42 | 12.56 | 12.58 | 12.62 |
| EPR | 12.15 | 12.35 | 12.36 | 12.52 | 12.56 | 12.57 |
| CEIL | 12.13 | 12.35 | 12.39 | 12.55 | 12.57 | 12.65 |
| **Diversity Pearson $r$** | $-0.869$ | $-0.979$ | $-0.865$ | $-0.799$ | $+0.685$ | $+0.503$ |

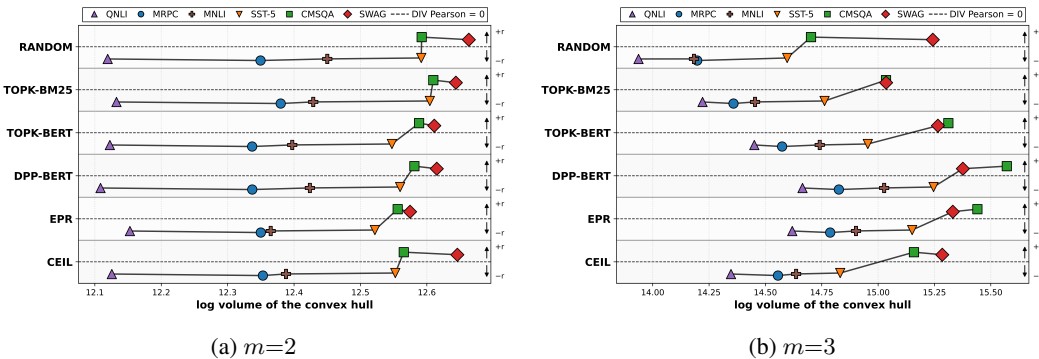

(a) $m=2$        (b) $m=3$

Figure 6: **PCA projections across dimensions.** Lower-dimensional projections ($m=2, 3$) retain the same qualitative ordering between hull log volume and the diversity–performance correlation as the $m=4$ analysis in the main text.

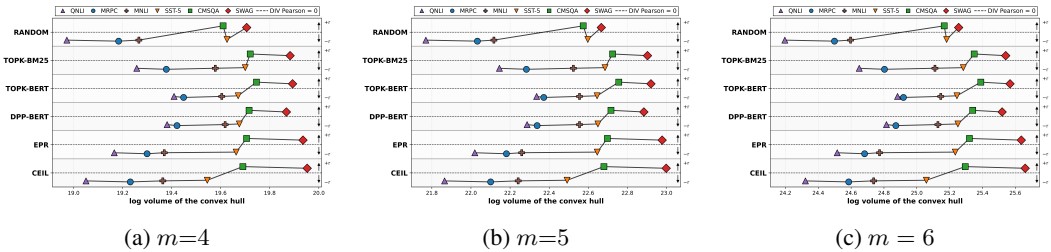

(a) $m=4$      (b) $m=5$      (c) $m=6$

Figure 7: **Rand–MC robustness.** Averaging the convex hull log $m$-volume over 100 random orthonormal projections preserves the dataset ordering for $m \in \{4, 5, 6\}$.

Table 7: PCA hull volume in 3 dimension (higher means a larger hull). The bottom row reports the dataset-wise diversity–performance correlation ($r$).

| | DATASET | | | | | |
|---|---|---|---|---|---|---|
| **METHOD** | QNLI | MRPC | MNLI | SST-5 | CMSQA | SWAG |
| RANDOM | 13.94 | 14.20 | 14.18 | 14.60 | 14.70 | 15.24 |
| TOPK-BM25 | 14.22 | 14.36 | 14.45 | 14.76 | 15.04 | 15.04 |
| TOPK-BERT | 14.45 | 14.57 | 14.74 | 14.95 | 15.31 | 15.27 |
| DPP-BERT | 14.66 | 14.83 | 15.03 | 15.25 | 15.57 | 15.38 |
| EPR | 14.62 | 14.79 | 14.90 | 15.15 | 15.44 | 15.33 |
| CEIL | 14.35 | 14.56 | 14.64 | 14.83 | 15.16 | 15.28 |
| **Diversity Pearson $r$** | $-0.869$ | $-0.979$ | $-0.865$ | $-0.799$ | $+0.685$ | $+0.503$ |

Table 8: PCA hull volume in 4 dimension (higher means a larger hull). The bottom row reports the dataset-wise diversity–performance correlation ($r$).

| | DATASET | | | | | |
|---|---|---|---|---|---|---|
| **METHOD** | QNLI | MRPC | MNLI | SST-5 | CMSQA | SWAG |
| RANDOM | 16.77 | 17.07 | 17.13 | 17.65 | 17.84 | 17.90 |
| TOPK-BM25 | 16.93 | 17.15 | 17.29 | 17.79 | 17.93 | 18.04 |
| TOPK-BERT | 16.98 | 17.15 | 17.29 | 17.63 | 17.91 | 18.13 |
| DPP-BERT | 16.93 | 17.13 | 17.27 | 17.69 | 17.88 | 18.10 |
| EPR | 16.82 | 17.07 | 17.08 | 17.68 | 17.80 | 18.06 |
| CEIL | 16.83 | 17.12 | 17.13 | 17.46 | 17.82 | 18.18 |
| **Diversity Pearson $r$** | $-0.869$ | $-0.979$ | $-0.865$ | $-0.799$ | $+0.685$ | $+0.503$ |

Table 9: Rand-mc hull volume in 4 dimension (higher means a larger hull). The bottom row reports the dataset-wise diversity–performance correlation ($r$).

| | DATASET | | | | | |
|---|---|---|---|---|---|---|
| METHOD | QNLI | MRPC | MNLI | SST-5 | CMSQA | SWAG |
| RANDOM | 18.97 | 19.18 | 19.27 | 19.63 | 19.61 | 19.71 |
| TOPK-BM25 | 19.26 | 19.38 | 19.58 | 19.70 | 19.72 | 19.88 |
| TOPK-BERT | 19.41 | 19.45 | 19.60 | 19.67 | 19.75 | 19.89 |
| DPP-BERT | 19.38 | 19.42 | 19.62 | 19.68 | 19.72 | 19.87 |
| EPR | 19.17 | 19.30 | 19.37 | 19.66 | 19.71 | 19.94 |
| CEIL | 19.05 | 19.23 | 19.36 | 19.55 | 19.69 | 19.95 |
| **Diversity Pearson** $r$ | $-0.869$ | $-0.979$ | $-0.865$ | $-0.799$ | $+0.685$ | $+0.503$ |

Table 10: Rand-mc hull volume in 5 dimension (higher means a larger hull). The bottom row reports the dataset-wise diversity–performance correlation ($r$).

| | DATASET | | | | | |
|---|---|---|---|---|---|---|
| METHOD | QNLI | MRPC | MNLI | SST-5 | CMSQA | SWAG |
| RANDOM | 21.77 | 22.03 | 22.12 | 22.60 | 22.57 | 22.67 |
| TOPK-BM25 | 22.15 | 22.28 | 22.52 | 22.69 | 22.72 | 22.90 |
| TOPK-BERT | 22.34 | 22.37 | 22.56 | 22.64 | 22.76 | 22.92 |
| DPP-BERT | 22.29 | 22.34 | 22.55 | 22.65 | 22.72 | 22.88 |
| EPR | 22.02 | 22.18 | 22.26 | 22.65 | 22.70 | 22.98 |
| CEIL | 21.87 | 22.10 | 22.24 | 22.49 | 22.68 | 23.00 |
| **Diversity Pearson** $r$ | $-0.869$ | $-0.979$ | $-0.865$ | $-0.799$ | $+0.685$ | $+0.503$ |

Table 11: Rand-mc hull volume in 6 dimension (higher means a larger hull). The bottom row reports the dataset-wise diversity–performance correlation ($r$).

| | DATASET | | | | | |
|---|---|---|---|---|---|---|
| METHOD | QNLI | MRPC | MNLI | SST-5 | CMSQA | SWAG |
| RANDOM | 24.20 | 24.50 | 24.60 | 25.18 | 25.17 | 25.25 |
| TOPK-BM25 | 24.65 | 24.80 | 25.11 | 25.28 | 25.35 | 25.54 |
| TOPK-BERT | 24.88 | 24.92 | 25.15 | 25.24 | 25.39 | 25.57 |
| DPP-BERT | 24.82 | 24.87 | 25.13 | 25.25 | 25.34 | 25.52 |
| EPR | 24.52 | 24.68 | 24.77 | 25.23 | 25.32 | 25.64 |
| CEIL | 24.32 | 24.59 | 24.74 | 25.06 | 25.30 | 25.66 |
| **Diversity Pearson** $r$ | $-0.869$ | $-0.979$ | $-0.865$ | $-0.799$ | $+0.685$ | $+0.503$ |

**Experimental procedure for Q3.** The following is our experimental procedure. (i) Prompts are entered and the CoE $H_P$ is obtained. (ii) We project onto the top singular directions with $m{=}4$ in the main text and report additional results at $m{=}2, 3$ in Fig. 6. (iii) For Rand-MC, we use $n_{\text{proj}}{=}100$ random orthonormal bases with $m \in \{4, 5, 6\}$ (Fig. 7). (iv) We report the *log $m$-volume*. Rankings are invariant under this monotone transform. (v) Per-dataset statistics are means over prompts.

**CoE volume on generation tasks.** As shown in Tab. 12, we also performed the PCA hull volume in 4 dimensions with the *random* sampler on the generation tasks. Compared with Tab. 8, the CoE volume of the generation tasks is consistently larger than that of all classification tasks.

# E  ADDITIONAL RESULTS

**Additional results for label-free cross-domain adaptation with LRPG.** Table 13 reports a CEIL retriever trained on **MRPC** and adapted to diverse targets using LRPG. As before, LRPG

Table 12: PCA hull volume in 4 dimensions (higher means a larger hull) for the generation task. Compared to the results obtained on the classification task (Tab. 8), the overall volume is larger.

| | DATASET | | | | | |
|---|---|---|---|---|---|---|
| **METHOD** | WebQs | GeoQ. | MTOP | NL2Bash | SMCal. | Break |
| RANDOM | 19.67 | 18.27 | 18.81 | 19.24 | 18.79 | 19.23 |

measures each target's CoE geometry and uses it to decide whether to increase or suppress diversity during sampling, without accessing target labels. In this setting learning-free baselines remain competitive under cross-domain shift. Even so, LRPG improves the learning-based sampler: CEIL+LRPG raises the overall average by $+1.97\%$ over CEIL, while the reversed policy CEIL+LRPG-REV degrades performance by $-3.40\%$. The contrast confirms that the direction of diversity adjustment matters and that CoE-guided control reliably benefits learning-based retrieval in another cross-domain scenario.

Table 13: Label-free cross-domain transfer to diverse target datasets. The retriever is trained on MRPC and evaluated on targets without access to target labels. The table highlights CEIL+LRPG, our CoE-based rule that decides whether to increase or suppress diversity for each target, and CEIL+LRPG-REV, which applies the opposite decision. We reports gains relative to CEIL.

| Method | Classification | | | | | Generation | | | | | | Avg. |
|---|---|---|---|---|---|---|---|---|---|---|---|---|
| | SST5 | QNLI | MNLI | CMSQA | SWAG | WebQs | GeoQ. | NL2Bash | Break | MTOP | SMCal. | |
| *Learning-free* | | | | | | | | | | | | |
| RANDOM | 32.97 | 56.84 | 38.85 | **59.79** | 67.46 | 4.92 | 33.93 | 14.81 | 1.83 | 6.67 | 3.84 | 29.26 |
| TOPK-BM25 | 37.24 | 64.62 | 45.30 | 54.87 | 68.37 | 16.63 | 62.86 | 34.48 | 25.41 | **54.09** | 44.42 | 46.21 |
| TOPK-BERT | 37.87 | 65.46 | 45.49 | 46.85 | 54.71 | **17.18** | **66.79** | 31.36 | 26.71 | 53.47 | **45.52** | 44.67 |
| DPP-BERT | 37.69 | 66.21 | 45.54 | 58.56 | **69.60** | 15.11 | 64.29 | **35.83** | 26.78 | 53.42 | 45.50 | **47.14** |
| *Learning-based* | | | | | | | | | | | | |
| EPR | 38.32 | 62.07 | 45.73 | 59.38 | 66.73 | 14.52 | 66.07 | 30.44 | 23.97 | 42.64 | 38.93 | 44.44 |
| CEIL | 30.88 | 55.46 | 42.03 | 57.25 | 67.57 | 10.04 | 58.57 | 20.13 | 14.91 | 38.93 | 29.31 | 38.64 |
| CEIL+LRPG | 32.97 | 55.35 | 41.41 | 57.90 | 67.15 | 12.40 | 66.43 | 27.28 | 16.07 | 39.82 | 29.89 | 40.61 |
| CEIL+LRPG-REV | 30.97 | 55.37 | 44.49 | 56.59 | 67.11 | 7.97 | 46.07 | 16.61 | 9.54 | 30.07 | 22.81 | 35.24 |
| GAIN OVER CEIL | +2.09 | −0.11 | −0.62 | +0.65 | −0.42 | +2.36 | +7.86 | +7.15 | +1.16 | +0.89 | +0.58 | +1.97 |
| GAIN OF REV OVER CEIL | 0.09 | −0.09 | 2.46 | −0.66 | −0.46 | −2.07 | −12.50 | −3.52 | −5.37 | −8.86 | −6.50 | −3.40 |

**Full results across targets and samplers.** Full cross-domain results with diversity-scale sweeps for both the learning-based retriever CEIL and the learning-free sampler DPP-BERT are shown in Tabs. 14 to 16. The CEIL tables include the effect of changing the diversity factor under two source settings, and the DPP-BERT table varies its diversity scale without training. Across all sweeps the direction predicted by CoE geometry holds. Targets with large CoE hull volume such as **SWAG** and **CMSQA** tend to improve as diversity is increased, while targets with small CoE volume such as **MRPC** and **SST-5** often improve when diversity is reduced. For generation tasks, performance generally rises when relevance is emphasized by lowering the diversity scale. These patterns match the main analysis in Fig. 3, indicating that CoE geometry reliably identifies how to adjust diversity for both learning-based and learning-free samplers in label-free cross-domain adaptation.

Table 14: Ablation of the diversity scale for CEIL in cross-domain adaptation. The retriever is trained on SST-5 and evaluated on each target. The diversity scale is shown in parentheses, ranging from 0.01 to 1.0, with 0.10 as the baseline.

| Source Dataset | SST-5 | | | | | | | | | | |
|---|---|---|---|---|---|---|---|---|---|---|---|
| Target Dataset | MRPC | QNLI | MNLI | CMSQA | SWAG | WebQs | GeoQ. | NL2Bash | Break | MTOP | SMCal. |
| *Learning-free* | | | | | | | | | | | |
| Random | 67.16 | 58.84 | 38.85 | 59.79 | 67.46 | 4.92 | 33.93 | 14.81 | 1.83 | 6.67 | 3.84 |
| TOPK-BM25 | 68.63 | 64.62 | 45.30 | 54.87 | 68.37 | 16.83 | 62.86 | 34.48 | 25.41 | 54.09 | 44.42 |
| TOPK-BERT | 72.79 | 65.46 | 45.49 | 46.85 | 54.71 | 17.18 | 66.79 | 31.36 | 26.71 | 53.47 | 45.52 |
| DPP-BERT | 70.83 | 66.21 | 45.54 | 58.56 | 69.60 | 15.11 | 64.29 | 35.83 | 26.78 | 53.42 | 45.50 |
| *Learning-based* | | | | | | | | | | | |
| EPR | 71.32 | 62.07 | 45.73 | 59.38 | 68.73 | 14.52 | 68.07 | 30.44 | 23.97 | 42.64 | 38.93 |
| Ceil (0.01) | 68.87 | 68.63 | 46.80 | 59.62 | 68.53 | 16.83 | 65.36 | 32.23 | 28.83 | 54.00 | 45.22 |
| Ceil (0.05) | 69.85 | 66.28 | 47.03 | 58.56 | 69.34 | 14.91 | 61.79 | 28.97 | 27.98 | 53.02 | 44.28 |
| Ceil (0.10 base) | 69.85 | 62.38 | 45.77 | 59.30 | 69.20 | 14.22 | 54.64 | 29.34 | 24.51 | 49.49 | 40.40 |
| Ceil (0.5) | 62.25 | 55.17 | 42.44 | 59.13 | 68.10 | 7.09 | 48.93 | 15.26 | 3.81 | 20.04 | 14.51 |
| Ceil (1.0) | 64.95 | 55.39 | 42.60 | 59.79 | 68.20 | 9.94 | 44.64 | 20.82 | 12.89 | 37.81 | 28.55 |

Table 15: Ablation of the diversity scale for CEIL in cross-domain adaptation. The retriever is trained on MRPC and evaluated on each target. The diversity scale is shown in parentheses, ranging from 0.01 to 1.0, with 0.10 as the baseline.

| Source Dataset | MRPC | | | | | | | | | | |
|---|---|---|---|---|---|---|---|---|---|---|---|
| Target Dataset | SST-5 | QNLI | MNLI | CMSQA | SWAG | WebQs | GeoQ. | NL2Bash | Break | MTOP | SMCal. |
| *Learning-free* | | | | | | | | | | | |
| Random | 32.97 | 56.84 | 38.85 | 59.79 | 67.46 | 4.92 | 33.93 | 14.81 | 1.83 | 6.67 | 3.84 |
| TOPK-BM25 | 37.24 | 64.62 | 45.30 | 54.87 | 68.37 | 16.63 | 62.86 | 34.48 | 25.41 | 54.09 | 44.42 |
| TOPK-BERT | 37.87 | 65.46 | 45.49 | 46.85 | 54.71 | 17.18 | 66.79 | 31.36 | 26.71 | 53.47 | 45.52 |
| DPP-BERT | 37.69 | 66.21 | 45.54 | 58.56 | 69.60 | 15.11 | 64.29 | 35.83 | 26.78 | 53.42 | 45.50 |
| *Learning-based* | | | | | | | | | | | |
| EPR | 38.32 | 62.07 | 45.73 | 59.38 | 66.73 | 14.52 | 66.07 | 30.44 | 23.97 | 42.64 | 38.93 |
| Ceil (0.01) | 32.97 | 55.35 | 41.41 | 56.59 | 67.11 | 12.40 | 66.43 | 27.28 | 16.07 | 39.82 | 29.89 |
| Ceil (0.05) | 32.61 | 54.71 | 42.04 | 57.74 | 67.74 | 9.94 | 63.57 | 25.95 | 14.39 | 39.24 | 29.32 |
| Ceil (0.10 base) | 30.88 | 55.46 | 42.03 | 57.25 | 67.57 | 10.04 | 58.57 | 20.13 | 14.91 | 38.93 | 29.31 |
| Ceil (0.5) | 29.97 | 54.97 | 43.33 | 57.90 | 67.38 | 7.68 | 46.79 | 16.61 | 9.81 | 29.40 | 22.49 |
| Ceil (1.0) | 30.97 | 55.37 | 44.49 | 57.90 | 67.15 | 7.97 | 46.07 | 16.61 | 9.54 | 30.07 | 22.81 |

Table 16: Ablation of the diversity scale for DPP-BERT in cross-domain adaptation. The diversity scale is shown in parentheses, ranging from 0.0025 to 1.0, with 0.05 as the baseline.

| Method | MRPC | SST-5 | QNLI | MNLI | CMSQA | SWAG | WebQs | GeoQ. | NL2Bash | Break | MTOP | SMCal. |
|---|---|---|---|---|---|---|---|---|---|---|---|---|
| DPP- BERT (0.0025) | 70.83 | 38.60 | 65.11 | 45.42 | 58.39 | 68.96 | 17.27 | 69.29 | 30.75 | 26.44 | 53.20 | 45.20 |
| DPP- BERT (0.005) | 70.34 | 40.33 | 64.47 | 44.88 | 58.15 | 69.13 | 17.47 | 67.14 | 33.31 | 26.48 | 53.11 | 45.39 |
| DPP- BERT (0.01) | 71.81 | 39.78 | 65.09 | 44.69 | 58.15 | 69.32 | 16.24 | 65.00 | 34.21 | 26.60 | 53.42 | 45.51 |
| DPP- BERT (0.025) | 72.06 | 38.24 | 65.44 | 45.37 | 57.90 | 69.54 | 15.99 | 64.64 | 32.72 | 26.65 | 53.06 | 45.56 |
| DPP- BERT (0.05 base) | 70.83 | 37.69 | 66.21 | 45.54 | 58.56 | 69.60 | 15.11 | 64.29 | 35.83 | 26.78 | 53.42 | 45.50 |
| DPP- BERT (0.1) | 69.85 | 39.60 | 65.81 | 46.36 | 58.31 | 69.66 | 14.57 | 57.14 | 33.11 | 25.91 | 52.93 | 45.08 |
| DPP- BERT (0.25) | 69.36 | 37.33 | 66.26 | 45.13 | 59.04 | 69.77 | 13.93 | 52.50 | 28.50 | 25.79 | 52.08 | 44.36 |
| DPP- BERT (0.5) | 69.85 | 36.33 | 67.58 | 45.16 | 58.89 | 69.77 | 13.83 | 48.93 | 28.93 | 25.21 | 51.86 | 44.46 |
| DPP- BERT (1.0) | 70.34 | 37.42 | 66.94 | 45.17 | 58.47 | 69.76 | 13.88 | 49.64 | 33.48 | 25.19 | 51.68 | 43.94 |

