# OpenReview forum: "Leveraging LLM Hidden-State Trajectories for Label-Free Sampler Adaptation"
_ICLR.cc/2026/Conference — ICLR 2026 Conference Withdrawn Submission_

### Official Review · Reviewer_P6P7 · 2025-10-30

**Soundness:** 3
**Presentation:** 3
**Contribution:** 3
**Rating:** 6
**Confidence:** 3

**Summary:**

This paper proposes a label-free method for adapting demonstration samplers in cross-domain in-context learning. The authors present Latent Reasoning Path Guidance (LRPG), which leverages the geometry of hidden-state trajectories, referred to as the Chain-of-Embedding (CoE), to infer whether to prioritize relevance or diversity in retrieval. The method operates without access to target-domain labels and can be applied on top of existing retrieval strategies. Experimental results demonstrate consistent improvements in ICL performance across a range of classification and generation tasks.

**Strengths:**

1. The paper provides a well-structured and systematic analysis of the relevance–diversity–performance correlation in in-context learning (ICL), clearly organized through targeted questions (Q1, Q2, Q3).
2. The investigation of the relationship between the Chain-of-Embedding (CoE) geometry and diversity is insightful and reveals a strong latent signal for retrieval preferences.
3. The paper conducts comprehensive benchmarking across both classification and generation tasks, offering a broader perspective on the effectiveness of the proposed method.
4. The use of the reversed LRPG (LRPG-REV) policy serves as a compelling ablation to validate the significance of the directionality signal derived from CoE statistics.

**Weaknesses:**

1. The experiments are conducted with a single backbone model (GPT-Neo 2.7B). It remains unclear whether the proposed findings and the effectiveness of LRPG generalize to other architectures or larger-scale models.
2. The method involves computing PCA hull volume from layer-wise hidden representations, which may introduce additional computational overhead. The paper lacks a discussion or analysis of the cost and feasibility of this computation, especially for larger models or real-time applications.

**Questions:**

1. In Figure 2, the authors state that EPR shows significant performance drops on SWAG and CMSQA under cross-domain transfer. However, the heatmap appears to show relatively high values for these domains. Since the figure shows "relative performance" normalized to 100% per source domain, could the authors clarify how to interpret these values and reconcile them with the reported drop?
2. As larger models (e.g., 7B+ scale) are increasingly common in ICL applications, has the proposed LRPG method been evaluated or tested for scalability on such models?

---

### Official Review · Reviewer_VB6E · 2025-11-01

**Soundness:** 2
**Presentation:** 2
**Contribution:** 2
**Rating:** 2
**Confidence:** 4

**Summary:**

The paper studies label-free cross-domain ICL and proposes using hidden-state trajectories of LLMs (chains of embeddings) to infer whether a target domain benefits more from relevance or diversity in demonstration selection. The key intuition is that the geometry of hidden-state activations (approximated via PCA hull volume) reflects the model’s reasoning dynamics and can guide retrieval-based samplers without labeled data in the target domain.

**Strengths:**

- The empirical observation that domains differ systematically in their preference for relevance vs. diversity is insightful and practically valuable.
- LRPG appears lightweight, easy to integrate, and computationally inexpensive.
- The study spans multiple datasets, including both classification and generation tasks, and provides ablations on CoE computation and PCA projection choices.

**Weaknesses:**

- The paper is difficult to follow due to poor organization and missing definitions. Key experimental details (models, datasets, samplers) appear late, forcing the reader to backtrack. Figures and references (e.g., Figure 1) often lack clear captions or context, and the Related Work section reads as a dense list rather than a synthesis.
- The evaluation is limited to a single model, leaving open whether results generalize across architectures. Because sampler behavior can depend strongly on embedding space structure, this raises concerns about robustness. The description of cross-domain evaluation (Figure 2) is confusing: values above 100 supposedly indicate improvement, yet those same cases are described as poor transfer (e.g., EPR on SWAG and CMSQA). This contradiction, together with the ambiguous explanation of how labels and demonstrations are used across source and target domains, raises concerns about interpretability. The paper describes a label-free cross-domain setup, yet it appears that source-domain labels are still required to train the sampler, while the target domain remains unlabeled. Moreover, it is unclear how the transfer operates if demonstrations are not directly used from the source domain; the method seems to rely only on query representations for adaptation. Without a clear description of what information is transferred and how supervision is restricted, it is difficult to assess the actual degree of label independence and the fairness of the comparison to other cross-domain or semi-supervised baselines.
- The PCA hull volume metric seems heuristic, with little justification for why it should correspond to diversity preference. Parts of the exposition rely on metaphorical phrasing ("preconditioning the reasoning space") rather than mechanistic explanation, reducing conceptual clarity.
- The quantitative results appear to partially contradict the claims. In Table 2, the reported gains for LRPG and its reversed variant are within noise margins, undermining the assertion that certain datasets are "diversity-beneficial." These inconsistencies echo my confusion around Figure 2’s interpretation.
- Figures 1 and 2 lack clear legends, labels, and consistent ordering. The color scales are poorly explained, making the heatmaps hard to interpret.
- The RW section is overly dense and somewhat repetitive, listing many retriever variants and heuristics without clearly organizing them by category or synthesizing how they relate to the proposed framework.

**Questions:**

1. Can you clarify the setup of label-free cross-domain ICL? Are source labels used for sampler training, and how are target demonstrations selected without any labels?
2. Could you clarify the interpretation of the values greater than 100 in Figure 2? According to the caption, they indicate improvement over the in-domain baseline, yet the text describes these same cases as poor transfer.
3. Can you provide a more formal explanation or theoretical intuition for the observed correlation between CoE volume (PCA hull size) and diversity preference? For example, does larger geometric dispersion in hidden-state trajectories indicate higher variance in task-relevant features, or does it reflect broader coverage of the input manifold?

---

### Official Review · Reviewer_eedp · 2025-11-01

**Soundness:** 3
**Presentation:** 3
**Contribution:** 3
**Rating:** 4
**Confidence:** 4

**Summary:**

This paper studies how to improve in-context learning under cross-domain shift by adapting which demonstrations are selected without using target labels. It argues that the balance between relevance and diversity is domain dependent, since some targets gain from diverse examples while others require strictly relevant ones. The method, called Latent Reasoning Path Guidance, reads the model’s hidden-state trajectory across layers to infer a label-free signal about the target domain. A broad trajectory suggests emphasizing diversity, while a narrow one suggests prioritizing relevance. The approach adjusts an existing retriever on the fly and shows consistent gains across text classification and question answering benchmarks.

**Strengths:**

- The paper pinpoints a real failure mode of fixed demonstration policies under cross-domain transfer and frames a concrete relevance versus diversity question tied to target properties.

 - The guidance rule is simple and interpretable, using the model’s own representation geometry as a label-free proxy for domain needs, so it can wrap existing retrievers without extra supervision.

- Results are consistent across multiple datasets, and the gains align with the central story that the optimal relevance versus diversity balance is domain dependent.

- Analysis goes beyond averages, including ablations and diagnostics that link trajectory breadth to retrieval choices, which helps the reader understand when the method is likely to help.

**Weaknesses:**

- Several explanations are terse and could be clearer, especially around Figure 2 and the cross-domain patterns that follow, as well as the motivation for using PCA hull volume as the signal. Clarifying these would make the core mechanism easier to trust (Question 1,2,3).

- LRPG communicates a binary decision to the sampler, diversity up or down, which limits finer control of the trade-off. A continuous or learned adjustment could better match intermediate regimes and reduce over or undercorrection.

- Against strong learning-free baselines such as DPP BERT the gains look marginal on average, which raises questions about where LRPG adds unique value beyond what a good diversity prior already captures.

**Questions:**

- In Figure 2 the in-domain story for SWAG and CMSQA is clear, yet when using these as sources the cross-domain drop is not as severe as expected. What properties of the targets or the sampling interact to prevent larger degradation in those transfers?

- What specific property of the PCA hull makes it a good indicator for preferring diversity over relevance? A short motivation that ties the metric to an intuitive picture of domain spread would help.

- Figure 4 is hard to read when printed, especially the labels. Can you enlarge or restructure the figure so the robustness evidence is visible at a glance?

- Can LRPG set the strength of the diversity term in a continuous way rather than a switch? If yes, how is the magnitude decided, if no, how should a practitioner tune the degree of diversity for a given target.

- DPP BERT often tracks the relevance versus diversity trade-off well and sometimes comes close to LRPG (Table 2). Is DPP already capturing much of the needed variability, or do you see specific cases where LRPG clearly outperforms due to its domain signal?

- ICLR requires disclosure of LLM usage in the submission. Where in the paper can this be found, or can you add a short statement to make the compliance explicit?

The contribution is clear on handling the relevance versus diversity trade off in cross domain settings. If the authors clarify the points raised above, I would be happy to raise my rating score.

---

### Note · Authors · 2025-11-12

**Comment:**

We thank all reviewers for their constructive feedback. After careful consideration, we have decided to withdraw our paper.

**Withdrawal Confirmation:**

I have read and agree with the venue's withdrawal policy on behalf of myself and my co-authors.